# The *Rm1* and *Rm2* Resistance Genes to Green Peach Aphid (*Myzus persicae*) Encode the Same TNL Proteins in Peach (*Prunus persica* L.)

**DOI:** 10.3390/genes13081489

**Published:** 2022-08-20

**Authors:** Henri Duval, Laure Heurtevin, Naïma Dlalah, Caroline Callot, Jacques Lagnel

**Affiliations:** 1INRAE, UR1052, Unité de Génétique et Amélioration des Fruits et Légumes (GAFL), F-84143 Montfavet, France; 2CNRGV, Université de Toulouse, INRAE, CNRS, F-31326 Castanet Tolosan, France

**Keywords:** peach, *Myzus persicae*, TIR-NBS-LRR (TNL), resistance gene, long read RNAseq

## Abstract

The green peach aphid (GPA), *Myzus persicae*, is an important pest of the peach crop. Three major dominant resistance genes have already been detected, *Rm1* in the Weeping Flower Peach (WFP) clone, *Rm2* in the Rubira clone, and *Rm3* in the Fen Shouxing clone. In this study, after NGS resequencing of WFP and Rubira, we found that their genomic sequences in the *Rm1* and *Rm2* region were similar but very different from that of the susceptible reference peach Lovell. We constructed a BAC library for the GPA-resistant WFP and screened four BAC clones to sequence the target region. The new sequence was 61.7 Kb longer than Lovell and was annotated with four different TIR_NBS_LRR genes. Among them, the *TNL1* gene was very overexpressed in WFP leaves 24 h after GPA infestation. This gene was also present and expressed in the Rubira clone and had the same sequence as the candidate *Rm3* gene, supporting the hypothesis that the three genes share the same origin. In addition, we identified a second TNL, *TNL2*, located at 35.4 Kb from *TNL1* and slightly overexpressed after GPA infestation. Kasp and size molecular markers were designed for use in marker-assisted selection and were validated in a peach segregating population.

## 1. Introduction

The identification of resistance genes to pests and pathogens in peach (*Prunus persica* L.) and other Prunus species is one of the main objectives of developing resistant peach and rootstock cultivars. Several mapping studies have been developed for locating *Prunus* resistance genes to pathogens [1,2] and only a few have been identified and functionally validated, such as the resistance genes to Root-knot nematodes *Ma* [3] and *RMja* [4] and the resistance gene to powdery mildew *Vr3* [5]. Now, with the development of reference genomes and new sequencing technologies (NGS, long read RNAseq, Pacbio), there are powerful tools to carry out detailed molecular studies leading to the identification of new genes and genetic markers. In peach, the peach reference genome V1 [6], available in 2010, was generated from the double haploid peach “Lovell” PLov2-2n, and a second version of the peach genome (Peach v2.0) [7] from the same genotype Lovell was achieved in 2016 (https://www.rosaceae.org/species/prunus_persica/genome_v2.0.a1 accessed on 1 June 2020). It allowed us to continue the research on the resistance genes to the green peach aphids (GPA), a major pest of peach causing leaf curling, stunted shoots, and weakening of the tree. In 1982, Massonié et al. [8] developed inoculation tests and identified two varieties, the peach Weeping Flower Peach (WFP) S2678 and the peach rootstock Rubira S4359, where aphid colonies did not develop and where resistance was associated with a necrotic reaction. The WFP S2678 clone was introduced to INRAE in 1961 from Clemson University with no information on its origin. It has red double flowers, a weeping growth habit, and is also resistant to the aphid *M. varians* Davids [9]. Rubira is a red leaf peach seedling rootstock released by INRAE from a Californian peach seed lot. 

Subsequently, several mapping studies were conducted by INRAE teams to identify the resistance genes of these two varieties, naming *Rm1* the gene present in the WFP clone [10,11] and *Rm2* the gene present in Rubira [12] and showing segregation of a single dominant gene. 

Notably, the WFP *Rm1* resistance gene to aphids causes strong antixenosis resistance [13]. This mechanism of resistance triggers hypersensitive-like necrotic reactions and protects the plant from aphid colonization. Based on this resistance mechanism, it can be expected that candidate genes belong to the family of the NLR (nucleotide-binding site-leucine-rich repeat). More recently, in China, a third single dominant GPA resistant gene, *Rm3*, was localized in the Chinese clone Fen Shouxing [14], and an NLR gene was identified as a candidate gene for GPA resistance [15]. As the two genes, *Rm2* and *Rm3*, trigger the same reaction as *Rm1* [14]), we hypothesized that these three genes probably belong to the same locus, but this hypothesis remains to be confirmed. 

Rough mapping placed the three genes at the end of chromosome 1, the *Rm1* locus in the genomic region (Pp01:43.62~46.50 Mb) [11], the *Rm2* locus in the genomic region (Pp01:45.08~46.23 Mb) [16], and the *Rm3* locus [17] within a 460-kb interval between the two SNP M1 SNP_Pp01_45.665389 and M2 SNP_Pp01_46.120950 (Figure 1a). In the hypothesis of a common origin and similarity of the three genes, the *Rm1* and *Rm2* genes should also be localized at the *Rm3* locus in the interval between the two markers M1 and M2, which is the smallest and is include in the genomic region encompassing the two other genes.

In this M1M2 interval, 53 genes (*Prupe.1G559400* to *Prupe.1G564500*) were annotated in the peach V2 reference genome and none of them as defense genes [9]. Four additional TNL genes (*ppa000596m*, *ppa017041m*, *ppa015956m* and *ppa018622m*) annotated on the peach V1.0 reference in the corresponding interval were reported on the Peach V2 by Van Ghelder and Esmenjaud [12] in a specific track on the GDR Jbrowse of the Peach V2 including all annotated TNL genes in peach. 

In this work, we decided to resequence the genomics region, including the M1M2 interval of the resistant genotypes WFP and Rubira, which harbor *Rm1* and *Rm2*, respectively, in order to look for additional TNL genes that may not be present in the peach reference V2. Using resequencing and expression data in WFP and Rubira, we identified four TNL genes in the targeted interval, two of which are overexpressed in aphid-infected plants. 

## 2. Materials and Methods

### 2.1. Plant Materials 

The trees of the two aphid-resistant peach clones, S2678 Weeping Flower Peach and S4359 Rubira, were maintained in the INRAE collection in 32l pots under insect-proof tunnels. The plants used for the GPA infestation tests were produced by grafting on the peach GF305 rootstock [8]. In this study, a population of 108 hybrids (Pamirskii × Rubira) × (Montclar × Nemared) PRMN previously described [18] were phenotyped for resistance to *M. persicae* and genotyped with specific molecular markers for the validation of the candidate genes. Thirty French peach accessions of the INRAE peach collections known as susceptible were genotyped with Kasp-specific markers (Appendix A). 

### 2.2. Phenotyping for Aphid Resistance

The test was carried out in a greenhouse under controlled conditions on young plants grown in small pots. After 3 months of growth, seedlings or grafted plants were infested by placing three adult apterous *M. persicae* on the terminal apex of each plant. The adult aphids were maintained and provided by the INRAE unit PSH (Plant and Horticultural System), which collected the aphid genotype from a peach tree in Avignon [13]. We observed the plants two weeks after infestation. Two classes of symptoms were observed and recorded. S: development of an aphid colony and leaf curling reactions, R: no aphid colony development, escape of aphids, and presence of necrotic spots (Figure 2).

### 2.3. Nucleic Acid Extraction and Genotyping

The genomic DNA of all tested plants was extracted using 100 mg of frozen young leaves. Each sample was ground with a mixer mill in 330 µL of extraction buffer (sorbitol 0.35 M, Tris 0.1 M, EDTA 5 mM, 4 mg of sodium metabisulfite), 330 µL of lysis buffer (Tris 0.2 M, EDTA 50 mM, NaCl 2 M, CTAB 2%), and 130 µL of sarkosyl 5%. After a chloroform-isoamylalcohol (24:1) procedure, precipitation with isopropanol and three ethanol washes, the DNA content was eluted in water and treated with RNAse. PCRs were carried out using the MyTaq DNA polymerase kit or the Expand long-range kit for longer fragments, following the manufacturer’s instructions.

### 2.4. NGS Genome Sequencing of the Peach Genotypes

The two resistant peaches, WFP and Rubira, and the parents of the PRMN population, Pamirskij, Montclar, and Nemared, were resequenced. The DNA was extracted from the leaves of the trees and maintained in pots under insect-proof tunnels. NGS resequencing was achieved using the platform MGX-Montpellier. Tagmentation and PCR amplifications were carried out using the Illumina Nextera DNA sample preparation kit following the supplier’s instructions. The validation of the genomics libraries was performed with DNA quantification using an Agilent high sensitivity chip and by qPCR. The cluster was carried out inside a flow cell using the Illumina cluster generation kit. Then, paired-end 125 sequencing was achieved using an Illumina Hiseq 2500 with a sequence by synthesis (SBS) technique. Pre-treatment of the data was performed using Illumina software HCS and RTA. The quality of the samples was analyzed with the software FastQC.

### 2.5. Genomic Library and Bacterial Artificial Chromosome (BAC) Sequencing

High molecular weight (HMW) genomic DNA was prepared from young frozen leaves picked from a potted tree kept in the dark for a week from the resistant peach WFP as described by Peterson [19] and Gonthier [20]. Agarose-embedded HMW DNA was partially digested with HindIII and subjected to two size selection steps by pulsed-field electrophoresis using a CHEF Mapper system. DNA was eluted, ligated into the pIndigoBAC-5 HindIII-Cloning Ready vector, and transformed into *Escherichia coli* electrocompetent cells. Pulsed-field migration programs electrophoresis buffer and ligation desalting conditions were performed based on the protocol of Challoub [21]. BAC clones were spotted on a nylon membrane, screened with radioactively labelled probes, and revealed by the high-density filter reader program. Eight labelled probes were designed using Primer 3 software from parts of sequences of WFP conserved with Peach V2 in the targeted genomic interval (Appendix A).

We verified the positive clones by real-time PCR using specific primers. In total, 2 µg of each individual BAC clone of interest was pooled for the construction of a SMRT^®^ library using the standard Pacific Biosciences preparation protocol for a 10 kb library with PacBio^®^ Barcoded Adapters.

The pool was then sequenced in one SMRT cell using P6 polymerase with C4 chemistry. Sequencing was performed on the PacBio RS II sequencer. After a demultiplexing step, the sequence assembly was performed following the HGAP PacBio workflow [22] and using SMRT^®^ Analysis software version 2.3 October 2014 for HGAP implementation BAC end sequences confirmed the position of selected clones on the peach genome. 

### 2.6. Construction and Sequencing Full-Length RNA-Seq Library

RNA extractions: We extracted total RNA from 2 apex leaf samples and 2 root samples using the NucleoSpin^®^ RNA Plant kit (Macherey Nagel). RNA concentration was measured using a Nanodrop^®^ by spectrophotometric absorbance at 260 and 280 nm. The quality of total RNA was evaluated using an Agilent 2100 Bioanalyzer (Agilent Technologies, Santa Clara, CA, USA). Only samples with RNA integrity number (RIN) ≥ 8 were used for deep sequencing.

Library construction and RNA sequencing: The complementary DNA was prepared from 500 ng of total RNA using the NEB Next^®^ Single Cell/Low Input cDNA Synthesis and Amplification Module kit. We used the SMRTbell^®^ Express Template Prep Kit 2.0 to prepare libraries and multiplex up to 2 samples. PacBio Sequel^®^ Systems was used to sequence transcriptomes. Full-length reads were selected and trimmed of 5′ and 3′ primers with specific adaptors (CGCACTCTGATATGTG for reverse primer of RNA apex, CTCACAGTCTGTGTGT for reverse primer of RNA roots). 

The raw SMRT sequencing data for this work was mapped onto the Peach V2 reference sequence, to which a new scaffold corresponding to the de novo sequences from the previously produced BAC libraries was added. We used the Iso-Seq 3 bioinformatics workflow in SMRT Link Version 7.0.1.66975 to produce full-length transcript sequences and splice variants of good quality, with no assembly required. The Iso-Seq analysis workflow starts with the generation of high-fidelity reads using the circular consensus sequencing (CCS) method and the required trimmed poly(A) tail. The minimum accepted subread length was 50 bp, and we added the isoseq-mode option.

### 2.7. Gene Expression (qPCR) 

Total RNA from apex leaves of WFP genotypes, from aphid-infected after 24 h and from healthy plants was extracted from frozen samples using the NucleoSpin^®^RNA Plant Kit (Macherey-Nagel, Dueren, Germany) according to the manufacturer’s instructions. RNA yield and purity were quantified using a Nanodrop^®^ by spectrophotometric absorbance and agarose gel electrophoresis. In total, 500 mg of cDNAs were synthesized using the Affinity Script Multiple Temperature cDNA Synthesis Kit (Agilent, Santa Clara, CA, USA), following the instructions provided. Expression analysis was performed by quantitative real-time PCR (qRT-PCR) using a “Stratagen MX Pro 3005” and Brilliant III SYBR Green qPCR Master Mix with Low ROX (Agilent Technologies). The reaction was prepared with each primer (0.5 µL, 10 µM), 2 µL of 1:10 diluted cDNA, RNase-free water (5 µL), and Brilliant III SYBR Green qPCR Master Mix with High ROX (7 µL) in a total volume of 15 µL. PCR conditions were 95 °C for 10 min, followed by 40 cycles of 95 °C for 30 s and 60 °C for 1 min. All qPCR reactions were normalized using the Ct value corresponding to the EiF4Ɣ peach gene (Pp_Eif4Ɣ) and the 60 S L13 ribosomal protein gene (Pp_RPL13). Each sample was measured three times with three technical replicates. The list of the designed primers for the qPCR is in Appendix A. 

### 2.8. NGS and Gene Sequence Analysis

The sequence alignment on the peach genome V2 was obtained using BWA (v 0.7.12-r1039) with the BWA-MEM algorithm for mapping. The read alignment and index files (BAM and BAI files), genomic feature files (GFF3) on the reference Peach V2, and de novo sequences were visualized with the Integrative Genomics Viewer (IGV) [23]. We used the CLC workbench (Qiagen, Hilden, Germany) for the sequence trimmings, translation proteins, gene annotations, and motif research. The software used for the gene prediction was FGENESH [24], with *Pp* as organism-specific gene-finding parameters (http://www.softberry.com) (accessed on 1 June 2020). Protein analysis and domain research was carried out with Interproscan (http://www.ebi.ac.uk/interpro/) (accessed on 1 June 2020).

## 3. Results

### 3.1. Reconstruction of the Genomic Region Encompassing Rm1 in WFP 

The Rm1 gene was previously located between the markers UDP022 and UDPAP467 and the Rm3 gene between the markers M1 SNP_Pp01:45.665389 and M2 SNP_Pp01_46.120950. To obtain the genomic sequence of WFP in this interval we resequenced WFP on an Illumina Hiseq 2500. We obtained 63.6 million short reads after including purifying filters for a depth of 42×. Only 2.4% of WFP reads were unmapped on the Peach V2. In the targeted M1M2 interval, there was deficient mapping to allow reconstruction of the complete WFP sequence in this interval. Indeed, in this interval, there were some parts of sequences conserved between WFP and Lovell, some dissimilar sequences detected when no WFP short-reads could align on the Lovell sequence and then the impossibility to reconstruct the WFP sequence.

To construct a physical map of the dissimilar region, we built a BAC library of the WFP accession with the aim of sequencing some BAC clones of this chromosome region. Our library contained 18,432 clones with an average insertion size of 119 kb and genome coverage of 7.6×. We screened 27 BAC clones, including the Rm1 regions, with the eight specific probes designed from the conserved sequences in the M1M2 interval. After aligning all the BAC ends on the Peach V2, we selected four BACS (20F07, 40H09, 4K22, and 41K15. See Figure 1b) and sequenced them using PacBio long-read sequencing. These BAC sequences covering the Rm1 region meet the quality criteria with a quality value of QV > 48, meaning high confidence in nucleotides. The fasta files were assembled by overlapping regions between the four BACs. With the mapping of the WFP NGS short-reads on the four BAC sequences, the few deletions or insertions in the overlapping BACs could be elucidated and resolved, and we obtained, between the two flanking markers M1 and M2, one unique consensus sequence “Rm1bac” of 517.362 kb (Appendix A). This sequence from WFP is 61.7 kb longer compared to the same sequence interval in the Lovell peach genome that was sequenced in the reference V2 (455.560 Kb), suggesting it could include additional genes. We decided to look for potential resistance genes in this locus. 

### 3.2. Structural Annotation of the TNL Genes in the Novel Rm1Bac Sequence from WFP 

In Peach V2, 53 genes (*Prupe.1G559400* to *Prupe.1G564500*) and 4 TNL genes (*ppa000596m*, *ppa017041m*, *ppa015956m* and *ppa018622m*) were annotated in the 460Kb M1M2 region (Figure 3a). To look for additional genes in the Rm1bac sequence, we used the predictive tool FGENESH and the prediction revealed 116 genes with 6 TNL genes, positioned as genes 17, 24, 56, 60, 92, and 93 (Figure 3b and Appendix A). Those TNL were compared with the four TNL annotated on the peach V2. *TNL1* and *TNL2* have 83.31 and 67.96 % nucleic identity with *ppa000596m*, *TNL3* 92.41% nucleic identity with *ppa017041m*, *TNL4* 99.49% nucleic identity with *ppa015956m*, *TNL5* and *TNL6* 79.82%, and 80.66% nucleic identity with *ppa018622m*. *TNL1*, *TNL2*, *TNL5*, and *TNL6* are, in a negative sense as the *ppa000596* and *ppa**018622m* genes, while *TNL3*, *TNL4*, *ppa**01704m1*, and *p**pa015956m* are positive genes.

We can conclude that *TNL3* close to *ppa017041m* and *TNL4* identical to *ppa015956m* are quite conserved in both genomes, while *TNL1* and especially *TNL2* are different from *ppa000596m*. *TNL5* and *TNL6*, different from *ppa018622m*, are less conserved even if they are in the same sense. Thus, we can determine that the four genes *TNL1*, *TNL2*, *TNL5*, and *TNL6*, which are present in the resistant clone WFP and not in the susceptible clone Lovel, could have a role in aphid resistance.

To see if these TNL genes were expressed and to validate their predicted intron/exon structure, we carried out a long-read RNAseq with iso-seq and PacBio technology. 

RNAseq data were mapped on the Rm1bac genomic sequence showing that only four TNL genes were expressed, *TNL1* (gene17), *TNL2* (gene24), *TNL3* (gene56), and *TNL6* (gene93) (Figure 3c and Appendix A). The structure of the expressed genes, including the number and location of exons, number and location of introns, CDS region, and location of promoter and terminal regions (PolyA), was exactly the same as the *FGENESH* prediction for the *TNL3* gene but differed slightly for the three other genes in exon five for *TNL1*, and in the exon seven for *TNL2* and *TNL6*. The structure of the gene *TNL1* consisted of a sequence of 5755 bp with six exons and five introns, while the three other genes had seven exons and six introns and are, respectively, 6630, 4512, and 4518 bp long. In the long-read transcripts of the four genes, we found a single open reading frame encoding four proteins of 1497, 1768, 1504, and 1506 amino acid residues (Figure 4).

Using Interproscan analysis of the protein sequences, we confirmed that the four genes belong to the family of TIR-NLR (TNL) genes with the presence of a Toll-interleukin-1 receptor domain in the N-terminal part of the protein and display a complete structure with full-length domains. The three first exons correspond to a specific protein domain. The first exon contains the typical TIR Domain motifs described for *Arabidopsis* by Meyers [26], TIR1, TIR3, and TIR5; the second exon contains the NB domain motifs P-loop, GLPL, and MHDV, and the third exon contains the motifs NLL1, LRR1a, and PL-1 described by Van Ghelder and Esmenjaud [25] (Appendix A). The four other exons contain LRR and post-LRR (Leucine-Rich Repeat) domains, but with one less post-LRR exon for *TNL1*.

Our results showed that four TNL genes are expressed, and among them, three genes, *TNL1*, *TNL2*, and *TNL6*, are very different from the four TNL of the susceptible reference peach Lovell. In order to search which genes are involved in the resistance, we looked to see if they were overexpressed in GPA-infested plants, in the hypothesis of overexpression of the resistant gene in the presence of the pathogen, as has been shown for *GmKR3*, a TIR–NLR gene, conferring resistance to multiple viruses in soybean [27]. 

### 3.3. Quantitative Expression Analysis of the Transcript TNL Genes 

We analyzed the relative normalized expression profiles of the four genes *TNL1*, *TNL2*, *TNL3*, and *TNL6* from WFP trees infested with aphids for 24 h and from non-infested trees, using three specific primers for every gene (Appendix A). The expression analysis of infected and uninfected tissues showed that *TNL1* was highly overexpressed during infestation with a relative expression (RE) of 47.15 and significantly differentially expressed between infected or non-infested plants (*p* > 0.01) (Figure 5). *TNL2* was slightly more expressed in infested plants (RE of 2.58) compared to non-infested plants (RE of 0.38), and in contrast, *TNL3* and *TNL6* were very weakly expressed in infected and uninfected plants with no statistical differences, with the RE of 0.02/0.17 and 0.001/0.004.

In the analysis, only the *TNL1* gene expression showed a much higher value in infected plants of the resistant genotype, making the aphid resistance-associated *TNL1* gene a strong candidate gene as the gene encoding the *Rm1* protein. On the contrary, TNL3 and TNL6 are expressed very weakly in control and infected leaves and are therefore unlikely to be involved in aphid resistance. For the *TNL2* gene, being also slightly more expressed in infected plants, we could not conclude whether it has a function in resistance or not.

### 3.4. Genomic Sequence of Rm2 in Rubira 

The *Rm2* gene of Rubira triggers the same type of reaction as the *Rm1* gene of WFP and is located in the same region. One hypothesis is that *Rm2* would be the same NLR gene as *Rm1*. To check that, we analyzed the *Rm2* locus in Rubira. The NGS Rubira short reads were aligned onto the WFP Rm1bac sequence showing that the Rubira sequence was identical to the WFP Rm1bac sequence in this region, with the same four TNL genes as shown on the screenshot of the IGV viewer for TNL1 (Appendix A) with the addition of short reads of the susceptible peach Montclar. We then investigated the expression of TNL1 by RT-PCR. Transcription of the *TNL1* gene of Rubira was confirmed by PCR of leaf cDNA from GPA-infected Rubira plants, using two specific primers pairs designed from the *TNL1* sequence (Figure 6). The expression of *TNL1* in WFP and Rubira confirmed that the *TNL1* gene is a good candidate for the *Rm2* gene in Rubira, as *Rm1* was in WFP.

### 3.5. Marker Development for Marker Assisted Selection

Since *TNL1* and, to a lesser extent, *TNL2* are strong candidates for GPA-resistant genes, we decided to design dominant intragenic markers to identify other resistant peach genotypes that carry the *TNL1* and *TNL2* genes. We selected one size marker (ZP185-F3) for *TNL1* and another (ZP186-F4) for *TNL2* with an amplicon size of 756 bp and 669 bp for genomic DNA and 602 pb and 515 bp for cDNA, respectively. The presence or the absence of the amplicon by PCR reveals whether the gene is present or not. The gel with the resistant genotypes (WFP, Rubira) and three susceptible genotypes (Pamir, Montclar, and Nemared) is represented in Figure 7. These markers were validated on the progeny of the double cross (Pamir × Rubira) × (Montclar × Nemared) (PRMN), including the resistant parent, Rubira. In the 108 hybrids of the PRMN population, 63 were genotyped and phenotyped as resistant, while 45 hybrids were genotyped and phenotyped as susceptible. 

We also developed Kasp markers with SNPs flanking the two genes *TNL1* and *TNL2*. The two Kasp markers that were the most efficient and replicable were SP949, located 18 kb before *TNL1*, and SP1022, located 20 kb after *TNL2* (Figure 7a). The Kasp marker genotyping of the 108 PRMN hybrids was perfectly consistent with the phenotyping. Likewise, the 30 susceptible French peach accessions of the INRAE collection were genotyped as susceptible with the two Kasp markers. These two types of markers are good markers which can be useful in peach breeding programs to integrate the two genes *TNL1* and *TNL2*.

## 4. Discussion

Genetic resistance to aphids in peach varieties is a preferable control method to chemical control with insecticides. For this reason, Massonié et al. [8] introduced a wide variety of peaches from different origins and developed inoculation tests to find resistant varieties. After identifying the Weeping Peach Varieties S2678 (WFP) and the rootstock Rubira, several mapping studies were conducted by INRAE teams to localize the region of the *Rm1* and *Rm2* genes. In another study [14], a third similar gene, *Rm3*, was localized in the same region in a narrower interval. In this work, we built a BAC library of the resistant variety WFP to obtain the genomic sequence of this new targeted interval and found that additional TNL genes typical of dominant resistances are present in this genotype and are different from those present in the susceptible peach reference Lovell. Long-read RNAseq analysis showed that four TNL genes (*TNL1*, *TNL2*, *TNL3*, and *TNL6*) were expressed, and another expression analysis by qPCR showed that the *TNL1* gene had a very high expression level in aphid-infected plants compared to the non-infested plants; *TNL2* was slightly more expressed and the other two genes were not overexpressed. In addition, with the NGS resequencing of the resistant Rubira, we could see that WFP and Rubira have the same TNL genes in this region with an identical genomic sequence, and we could confirm that *TNL1* and *TNL2* were well expressed in Rubira as in WFP. 

We hypothesized that the three genes *Rm1*, *Rm2*, and *Rm3*, being localized in the same region, were closely related. This was confirmed by the latest work on *Rm3* by Pan et al. [15], who found a candidate gene for the aphid-resistant gene *Rm3* in peach “Fen Shouxing” that provides the same necrotic reaction as *Rm1* and *Rm2* when infected with *M. persicae*. They used the same methodology as ours to identify the *Rm3* gene using a BAC library and PacBio sequencing. They also found two new NLR genes, *NLR1* and *NLR2*, and identified *NRL1* as a strong candidate gene for *Rm3*. The comparison of the genomic sequence and transcripts of *TNL1* and *NRL1* shows that they are identical. They did not publish the sequence of their second gene, *NLR2*, but *TNL2* and *NRL2* could be different because *NRL2* is oriented in the positive strand while our *TNL2* gene is oriented in the negative strand; *NLR2* was not overexpressed in the infested plants, while *TNL2* was slightly overexpressed. Our results and those obtained on the *Rm3* gene support the hypothesis that the three loci have the same origin and the *TNL1* gene (=*NLR1*) is the same candidate gene for *Rm1*, *Rm2*, and *Rm3*. The role of *TNL2* is less clear, and it has either no action in GPA resistance or perhaps an action of regulation. 

Sauge et al. [28] found a slower effect of antixenosis in Rubira than in WFP. In their experiments, they counted the percentage of aphids remaining on plants after placement on the plant and in their 2006 experiments (10 aphids by experiment), after 96 h, all aphids escaped from WFP and 75% for Rubira. However, in the data of the same experiment in 2002 [29] (40 aphids by experiment), they observed that after 96 h, the aphids were almost all gone from Rubira plants, and they did not test WFP. Despite this slight difference in the escape dynamic of aphids, we showed that the two accessions carry the same *TNL1* gene, suggesting that the observed difference may be linked to the level of expression of the gene *TNL1*, regulated by another gene such as *TNL2*, which has been shown for some resistances mediated by NLR pairs [30]. Another explanation may be the role of the genetic background [31], which could have an influence on the expression level for plant viruses, where the genetic background may impair effector-triggered dominant resistances at several stages by modifying the NLR response pathway.

Only two major aphid resistance genes have been cloned so far, the melon *Vat* gene, which confers resistance to the melon–cotton aphid *Aphis gossypii* [32] and the *Mi-1* gene, which confers resistance to the potato aphid *Macrosiphum euphorbiae* [33]. The *Vat* gene also confers resistance to some viruses when they are transmitted by *A. gossypii* [34], and the gene *Mi-1* also confers resistance to root-knot nematodes [35] and other insects, psyllids and whiteflies [36]. These two genes belong to the Coil–coil (CC)-NLR family, encoding a protein with a coiled-coil domain in the N-terminal extremity of the NBS region, while the genes *TNL1* and *TNL2* are resistance genes of the sub-family Toll/interleukin-1 receptor (TIR)-NLR. This family is thought to have appeared earlier than the CNL family [37], and many TNL genes control diverse plant pathogens such as viruses, bacteria [38], nematodes [3], or fungi. Rubira is also known as a peach rootstock for its tolerance against *Pseudomonas syringae* [39], and it would be interesting to check if there is an effect of the *TNL1* gene on tolerance to the bacterial canker.

In this study, we identified the gene *TNL1* as a major candidate gene for the two known aphid resistance genes, *Rm1* in WFP and *Rm2* in Rubira, identical to the NRL1 candidate gene *Rm3*. A second gene, *TNL2*, located at 35 kb of *TNL1*, was shown to also be weakly expressed in WFP, but its role in aphid resistance is still unclear. It would be necessary to create a recombinant peach hybrid bearing the *TNL1* gene and not the *TNL2* gene to know the role of each one. One other possibility would be to clone the *TNL1* gene and transform a susceptible peach with a *TNL1* construction. In vitro peach regeneration and genetic transfer with *Agrobacterium* is difficult, but the first transgenic peach plant carrying the *cry1Ab* gene for insect resistance was successfully produced recently [40]. 

Using the sequence of the two genes and the region of the locus, we developed specific Kasp markers and intragenic markers to integrate these major aphid resistance genes in new peach rootstocks and peach varieties, which will require several backcrosses to improve the poor fruit quality of the two peach varieties WFP and Rubira. These markers would be indispensable if we want to cumulate aphid-resistant QTLs such as those detected in the wild peach, *Prunus davidiana* [41], with major resistance genes. 

## Figures and Tables

**Figure 1 genes-13-01489-f001:**
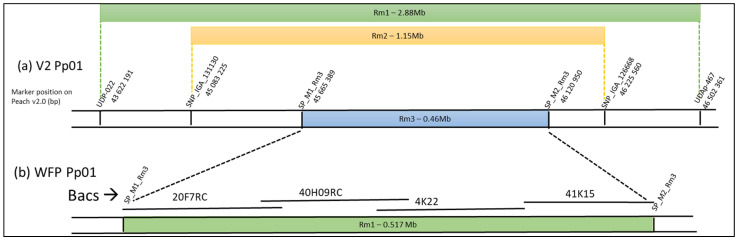
(**a**) Schemes of the genomic region identified by fine mapping for the *Rm1* [11], *Rm2* [16], and *Rm3* [17] genes. The position on the chromosome Pp01 peach V2 of the SSR and SNP markers framing the region are indicated. (**b**) Position of the four sequenced BAC clones screened from WFP leaves DNA.

**Figure 2 genes-13-01489-f002:**
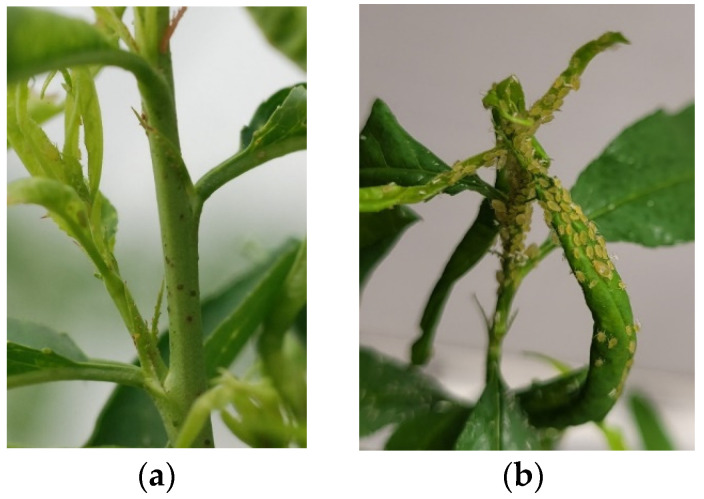
(**a**) Phenotype of the WFP resistant plant after infestation with GPA with the escape of aphids and presence of necrotic spots. (**b**) GPA colony development on a susceptible hybrid.

**Figure 3 genes-13-01489-f003:**
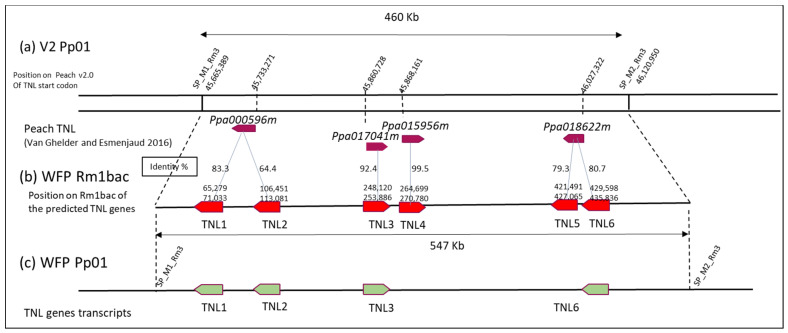
TNL genes on the Pp01 M1M2 interval: (**a**) on Peach V2.0 with the 4 TNL genes located by Van Ghelder and Esmenjaud [25]; (**b**) FGENESH prediction of TNL genes on the WFP genomic sequence (**c**) Expressed transcripts identified in the WFP long-read RNAseq.

**Figure 4 genes-13-01489-f004:**
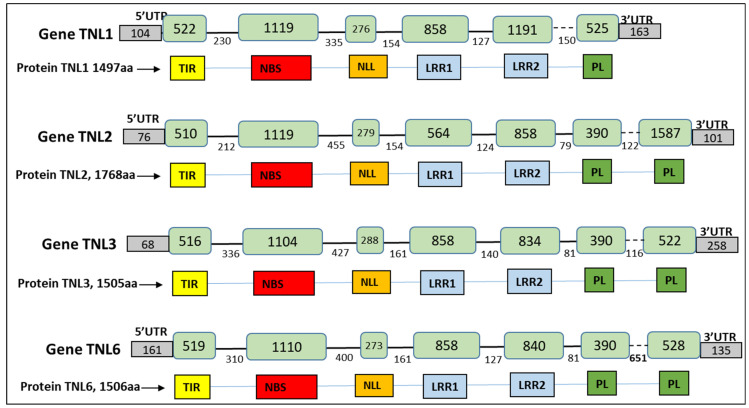
Structure of the four genes *TNL1*, *TNL2*, *TNL3*, and *TNL6*: size (bp) of exons, introns, and UTR, predicted domains of the TNL protein. Exons are represented in green boxes, introns by a black lane. The last intron in the dotted line contains one SSR (single sequence repeat).

**Figure 5 genes-13-01489-f005:**
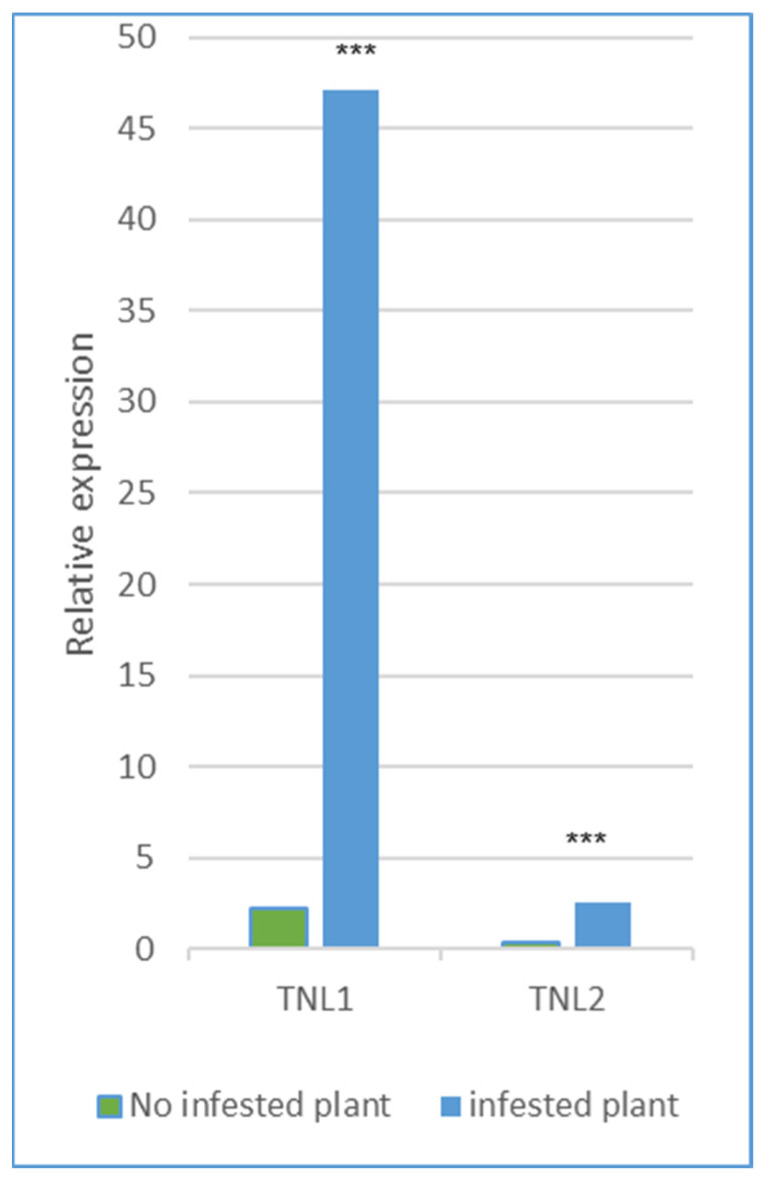
Expression profiling of the two genes, TNL1 and TNL2, without aphid infestation or after green peach aphid infestation in the resistant peach Weeping Flower (WFP). Data represented the mean of three replicates. *** Significative differences between no infested and infested.

**Figure 6 genes-13-01489-f006:**
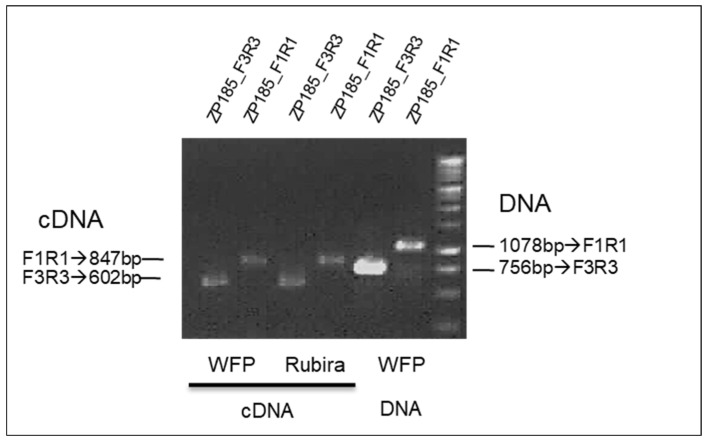
Genotyping with the two primers ZP185F1R1 and ZP185F3R3 specific of the *TNL1* gene, of the Rubira cDNA and Weeping flower peach (WFP) cDNA and DNA.

**Figure 7 genes-13-01489-f007:**
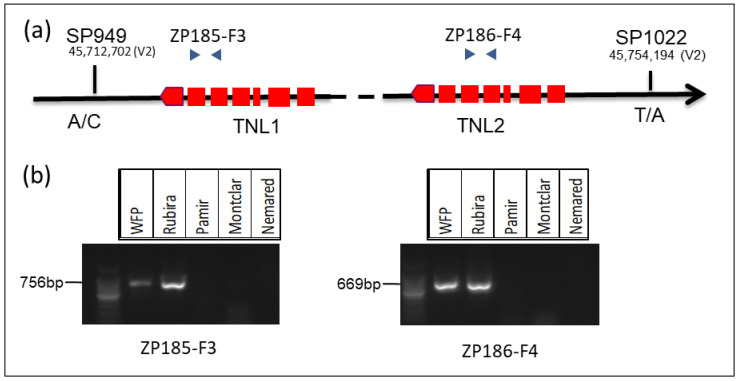
Genotyping for the *TNL1* and *TNL2* genes of resistant and susceptible peach parents. (**a**) Position of the SNP and specific size markers (**b**) genotyping with the size markers ZP185-F3 and ZP186-F4; the presence of amplicon for resistant and absence of amplicon for susceptible genotypes.

## Data Availability

Sequencing data for genome assemblies and contigs sequences were deposited in the NCBI database under the following numbers: Bioprojects PRJNA854381 and Biosample SAMN29438676 for Weeping Flower Peach, BioProject PRJNA854381 and Biosample SAMN29438677 for Rubira. All other relevant data are available on reasonable request to the Corresponding Author.

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
