# Peer review of "The Rm1 and Rm2 Resistance Genes to Green Peach Aphid (Myzus persicae) Encode the Same TNL Proteins in Peach (Prunus persica L.)"

_genes, 2022, doi:10.3390/genes13081489_

Round 1

Reviewer 1 Report

The authors combined bioinformatics and experimental methods to identify the aphid resistance genes in green peach. The related genes identified by bioinformatics analysis from sequencing were proved by the experiment. This study is a good work and will provide a good model for the study of biomarker discovery in plant.

Author Response

Please see the attachment with some english language corrections .

Reviewer 2 Report

Comments to the manuscript genes-1839843 intended as a research article in Genes entitled “The Rm1 and Rm2 resistance genes to green peach aphid (Myzus persicae) encodes the same TNL proteins in peach (Prunus persica L.)” by Henri Duval, Laure Heurtevin, Naima Dlalah, Caroline Callot and Jacques Lagnel.

They present a very interesting and concise manuscript on the genetic background on resistance to aphids in peach.

A project with a very nice experimental design. The manuscript has a good, comprehensive and thorough introduction to the topic. The methods used, the number of replicates and the analyses performed seem appropriate and solid. The results are clear and concise presented. The comprehensive discussion is covering the findings and topic nicely. This manuscript is a continued effort to identify and describe aphid resistance genes in peach.

 µl should be µL

 There are several minor language mistakes e.g. Ln310: reconsider using “slowly”, Ln317: change “than” to “as”, ln320: change “shot screen” to “screenshot”, ln372: change “did” to “made”.

Author Response

Please see the attachment with the correction of minor spells
